# Factors Contributing to Citizens’ Participation in COVID-19 Prevention and Control in China: An Integrated Model Based on Theory of Planned Behavior, Norm Activation Model, and Political Opportunity Structure Theory

**DOI:** 10.3390/ijerph192315794

**Published:** 2022-11-27

**Authors:** Xiaojie Zhang, Lili Wang

**Affiliations:** 1Department of Public Administration, School of Humanities & Law, Northeastern University, Shenyang 110169, China; 2Party School of Weihai Municipal Committee of Communist Party of China, Weihai 264213, China

**Keywords:** citizen participation, COVID-19, theory of planned behavior, norm activation model, political opportunity structure, openness to public participation, determinants

## Abstract

Chinese citizens’ participation in COVID-19 prevention and control has made great contributions to the successful fight against the pandemic. The factors that have impacted citizens’ participation have rarely been reported based on both social–psychological and political environmental theories. This paper presented a study to explore the determinants of Chinese citizens’ participation in COVID-19 prevention and control based on a combined model of the theory of planned behavior, the norm activation model, and political opportunity structure theory. A dataset involving 463 respondents from Harbin in northeast China was acquired and analyzed. The results showed that the comprehensive model explained 62.9% of the total variance in citizens’ participation behavior. The openness to public participation not only significantly directly influenced citizens’ participation but also indirectly affected participation behaviors through attitude and perceived behavioral control, both of which were important mediators and had the greatest overall impacts. The awareness of consequences and subjective norms were crucial antecedents to the activation of other influencing factors. Personal norms indirectly affected participation behavior through the mediation of attitude. The empirical results showed the comprehensiveness, effectiveness, and high explanatory power of the postulated model. The study also provides both new theoretical perspectives for explaining public participation and useful practical implications for future policy development in promoting citizens’ participation in public health emergency management.

## 1. Introduction

The coronavirus disease 2019 (hereafter COVID-19) that broke out at the end of 2019 in Wuhan, the capital city of Hubei province in central China, spread rapidly across the world, causing a global pandemic and leading to hundreds of thousands of deaths [1]. As of 12 November 2022, the number of global confirmed COVID-19 cases had reached 630.83 million, with 6.58 million confirmed deaths [2]. In response to the serious threat brought by the virus, various government efforts have been made by countries around the world to combat the pandemic [3]. China was the first country to implement preventive policies [4], and it has efficiently and successfully adopted a variety of measures to control and prevent the spread of the epidemic, thus providing rich experiences and lessons for other countries [5]. These measures include locking down cities or communities with a severe epidemic, suspending work, businesses and schools, sealing up foods and related items contaminated by the virus, quarantining the confirmed cases and their close contacts, providing free COVID-19 vaccinations and free medical treatment for people infected with COVID-19, etc. Due to the effective enforcement of these prevention and control measures, the total number of confirmed cases in China was 8.73 million, with less than 30 thousand confirmed deaths as of 12 November 2022 [6]. This is an extraordinary achievement compared to the large population in China.

The effectiveness of all these measures depends on citizens’ support and participation [7]. Chinese citizens have widely and deeply involved themselves in the collaboration with the government to fight against the epidemic, contributing to the great success of COVID-19 prevention and control [8]. Citizens’ participation in COVID-19 prevention and control in China mainly refers to their compliance with the measures enforced by the government, containing their preventive behaviors or protective behaviors. Specifically, citizens’ participatory behaviors include staying at home as required, wearing masks outside the household, measuring body temperature and scanning a health code in crowded places, maintaining social distance, reducing gatherings, washing hands frequently, performing nucleic acid tests regularly, reporting itinerary information to the community, and providing volunteer services. As such, the factors that have determined Chinese citizens’ high participation in the battle against the epidemic has become a hot research question.

Previous studies have addressed factors that contribute to citizens’ participation in COVID-19 prevention and control in both China and other countries, mainly based on the theory of planned behavior (hereafter TPB) or its extensions, the norm activation model (hereafter NAM) and social capital theory. For example, Das et al. [9] and Fischer and Karl [10] applied the TPB to investigate the determinants of COVID-19-relevant behaviors and found significant impacts of the three constructs of TPB. Irfan et al. [11] and Dutta et al. [12] used an expanded framework of the TPB to assess public willingness to comply with COVID-19 preventive measures. Their empirical results not only revealed significant effects of the TPB constructs, but also demonstrated the influences of other predictors, including the perceived risk of the pandemic, the perceived benefits and costs of preventive behaviors, and personal innovativeness. In addition to the direct impact of the three components of the TPB, Frounfelker et al. [13] and Li et al. [14] also found mediating roles of the three constructs in interpreting individuals’ preventive intentions or behaviors against COVID-19. In contrast to the above studies, Pan and Liu [15] and Ahmad et al. [16] uncovered insignificant effects of perceived behavioral control on individuals’ intentions to accept COVID-19 epidemic prevention measures. Patwary et al.’s [17] research showed no significant effects of attitude or subjective norms on people’s intention to be vaccinated against COVID-19. Shmueli found that only the subjective norms had a significantly positive impacts on respondents’ vaccination intentions [18].

A few studies adopted the NAM framework to examine various factors that impacted individuals’ involvement in the prevention and control of COVID-19. Chi et al.’s study [19] verified that personal norms efficiently affected festival travelers’ protective behaviors against COVID-19. An awareness of COVID-19 indirectly influenced travelers’ protective intentions through the sequentially mediating roles of ascription of responsibility, positive or negative anticipated feelings, and personal moral norms. However, the analysis by Ahmad et al. showed no contributions of moral norms to individuals’ intentions toward COVID-19 prevention in China [16].

A couple of other studies have integrated the TPB and the NAM to investigate public participation in the fight against the COVID-19 pandemic and demonstrated different impacts of the components in the two theories and their complicated relationships in explaining individuals’ COVID-19 prevention-relevant behaviors [20,21]. There are also some studies analyzing the antecedents of citizens’ participation in preventing and controlling the COVID-19 pandemic from other theoretical perspectives, including social capital theory [12,22], the individual–psychology–environment model [23], the knowledge–attitudes–practices model [24], and the health belief model [17,18]. In addition, several studies have focused on the roles of media, news consumption, civic talk, and political antecedents in predicting public engagement during the COVID-19 crisis [25,26].

As can be seen, most of the existing studies applied the TPB or its extension to analyze the factors contributing to citizens’ participation in COVID-19 prevention and control. Some studies adopted other theoretical frameworks, but only very few studies use the NAM or its integration with the TPB to explore the determinants of civic participation against the background of the COVID-19 pandemic. Furthermore, there are no studies addressing this research question based on both the TPB and the NAM in the Chinese context. In addition, the external environment [18], especially the political environment, plays a vital role in public participation [27]. However, previous research examining the impact of the external environment on citizens’ COVID-19 preventative intentions/behavior are only empirical analyses and not grounded in any established theories, leading to less theoretical contributions.

The current study is aimed to identify the important factors that impact citizens’ participation in COVID-19 prevention and control in China, based on an integrative theoretical model of the TPB, the NAM, and political opportunity structure theory. As such, we hope to inform policymakers in stimulating citizen participation in the context of a future pandemic or other public health emergencies.

## 2. Theoretical Framework and Research Hypotheses

### 2.1. The Theory of Planned Behavior

TPB was constructed to explain the factors affecting a person’s choice to perform or not perform a specific behavior or intention from the view of rational self-interest. It has been successfully applied to explain all kinds of self-interested as well as pro-social behaviors, such as environmental protection [28,29,30], voting behavior in general elections [31], healthcare and drug using [32,33], technology adoption [34], and so on. All these studies have proved the effectiveness and predictability of the TPB. Citizens’ participation behavior in COVID-19 can be regarded as not only a kind of self-interested behavior but also as a typical kind of pro-social behavior, because it may generate great values not only for individuals but also for the whole society. Hence, it is appropriate to use the TPB to explore why citizens participated in prevention measures since the outbreak of COVID-19.

According to Ajzen [35], attitudes towards behavior (ATT), subjective norms (SN), and perceived behavior control (PBC) are the main determinants to predict a person’s behavior or intention. ATT refers to one’s positive or negative appraisal of a specific behavior. Generally speaking, if a person holds positive ideas towards a behavior, such as the notion that the behavior is meaningful and valuable, he or she is more likely to perform the behavior. SN reflects the social pressure that individuals have perceived from important others such as families, friends, colleagues or leaders, and so on. Specifically speaking, a person is more likely to perform a behavior if important others relevant to him or her give encouragement or approval. PBC refers to a person’s perceived ease or difficulty in executing a specific behavior. In other words, it is an evaluation of one’s own internal conditions and ability to perform a certain behavior. In this study, PBC refers to citizens’ personal beliefs about how easy or difficult it is to participate in the prevention and control of COVID-19.

With the wide application of the TPB in interpreting various behaviors, relationships between the three constructs, including ATT, SN, and PBC, have also been uncovered. A lot of studies have confirmed the indirect link between SN and targeted behavior or intention by way of ATT and PBC [20,36,37,38,39,40]. Meanwhile, previous studies have also proved the relationship between ATT and PBC [38,41]. Therefore, the following hypotheses were proposed:

**Hypothesis 1 (H1).** *ATT is positively correlated with citizens’ participatory behaviors*.

**Hypothesis 2 (H2).** *PBC is positively correlated with citizens’ participatory behaviors*.

**Hypothesis 3 (H3).** *ATT is positively correlated with PBC*.

**Hypothesis 4 (H4).** *SN is positively correlated with ATT*.

**Hypothesis 5 (H5).** *SN is positively correlated with PBC*.

### 2.2. The Norm Activation Model

The NAM was first put forward by Schwartz to explore the factors that affect individuals’ behaviors from the perspective of altruism [42]. The NAM has been extensively used in explaining pro-social behaviors, including helping behavior [43], blood donation behavior, crowdfunding donation behavior [44,45], and environmental-friendly behavior [46,47,48,49,50].

Specifically, the NAM concentrates more on how individuals’ internal sense of morality comes into play in behavioral choice. The model has three key components, which are personal norms (PN), the awareness of consequences (AC), and the ascription of responsibility (AR). PN, also called moral norms, refers to an individual’s perceived moral obligation to perform or refrain a specific behavior or not [51]. AC involves whether an individual can realize the positive or negative consequences towards others caused by his or her specific behavior [40]. AR indicates whether an individual would attribute the consequence of the behavior to himself or herself.

With regard to the relationships between the NAM constructs, PN is considered to be directly related to the individual’s behavior or intention, and AC and AR are seen as the triggers for PN, which means that AC and AR are the prerequisites that activate an individual’s perception of moral obligation [39,52,53]. However, it should be highly emphasized that the application of NAM in different studies has been complex; especially when scholars incorporated the relevant variables in the NAM into the TPB model, some variables were abandoned [54]. For example, when studying residents’ garbage-sorting behavior, Goh et al. did not include AR into the analytical model and finally demonstrated the strong relationship between AC and PN [55], and some other empirical studies from China also revealed the same finding [56,57]. As a result, when combining NAM and TPB, we did not include AR in the integrated model based on previous studies.

Since AC reflects an individual’s belief about a specific behavior, and the belief is closely associated with ATT [58], researchers have considered that if an individual has a high level of AC, he or she would develop a more positive attitude toward implementing a specific behavior [59,60]. Aside from this, the correlation between PN and ATT has also been demonstrated [61,62]. In addition, relevant studies have also confirmed the connection between SN and PN and illustrated the prerequisite role of SN [38,63,64]. Specifically, SN refers to important others’ beliefs toward a definite behavior and the willingness to comply with important others, which is an important source of an individual’s internal moral sense. Furthermore, one can verify the social correctness of specific behaviors and judge whether their beliefs are right or wrong in terms of morality through SN. Hence, according to the original NAM assumptions and the empirical results in previous studies, we proposed the following research hypotheses:

**Hypothesis 6 (H6).** *AC is positively correlated with PN*.

**Hypothesis 7 (H7).** *AC is positively correlated with ATT*.

**Hypothesis 8 (H8).** *PN is positively correlated with ATT*.

**Hypothesis 9 (H9).** *SN is positively correlated with PN*.

### 2.3. Political Opportunity Structure Theory

Political opportunity structure (hereafter POS) theory is a theory of social movement grounded in political sociology. Its basic premise is that exogenous environmental factors, such as outside political opportunities or political threats, enhance or inhibit prospects for political activities [65] by opening avenues or imposing certain constraints. Eisinger was the first scholar who explicitly adopted the political opportunity framework. He defined political opportunity as the openness of governments that obstructs or facilitates citizens’ activities in pursuit of political goals, including impacting policy formation and decision making [66]. Since Eisinger initiated the term political opportunity, the POS theory has been utilized to explain a variety of citizens’ political activities, from the emergence and mobilization of social movements [67,68,69,70,71,72,73] to the strategies of activism [74,75,76] and public participation, including political participation [77] and environmental participation [27], demonstrating the strong explanatory power of the POS theory.

With the wide application of the theory, Eisinger’s original definition of POS has also been largely extended, and many new contents of essential political opportunity components have been included [78]. Kitschelt added state capacity to the openness of government [79]. Meyer and Minkoff classified POS into general and issue-specific POS, the former of which refers to general openness in the polity, and the latter refers to openness toward particular constituencies [65]. Other scholars have offered conceptualizations that employed three [80,81,82,83] and five components of political opportunity [84]. Although authors’ conceptions of the dimensions of political opportunity are various, they have overlapping concerns. The mostly overlapped dimension is openness, which was termed as access to the party system by Rucht [83], openness or closure of the polity by Sidney [84], and openness or closure of the institutionalized political system by McAdam et al. [82]. This indicates that openness, which was initially proposed by Eisinger, is a very important dimension of political opportunity [66].

Since POS theory is a good theoretical tool to interpret exogenous factors that influence citizens’ various political actions including participation behaviors, this study applies this theory to help to better explain the political determinants of citizens’ participation in COVID-19 prevention and control in China. As the openness of the political system is the initially raised and most frequently used dimension of the POS theory, and the extent of access to the political system is very important for public participation [85], this study proposed the following hypothesis:

**Hypothesis 10 (H10).** *The openness to public participation (hereafter OPP) is positively correlated with citizens’ participatory behaviors*.

Prior research has revealed significant relationships between the TPB constructs and the exogenous environmental factors in predicting citizens’ participation in disease prevention and control. For example, Li et al. found that the institutional climate with formal policies and procedures regarding COVID-19 prevention and control significantly influenced attitudes and perceived behavioral control [14]. Siuki et al.’s empirical results demonstrated that health education interventions concerning disease prevention were significantly related with attitudes and perceived behavioral control [86]. Based on the existing empirical evidence, we argue that the OPP affects citizens’ attitudes and perceived behavioral control towards COVID-19 prevention and control. Hence, we put forward the following two hypotheses:

**Hypothesis 11 (H11).** *OPP is positively correlated with ATT*.

**Hypothesis 12 (H12).** *OPP is positively correlated with PBC*.

### 2.4. Integrated Research Model

On the basis of the TPB, the NAM, POS theory, and the empirical evidence provided in the extant research mentioned above, an integrated analytical model was proposed for evaluating the roles of ATT, PBC, SN, AC, PN, and OPP in citizens’ participation behaviors against the COVID-19 pandemic in China (see Figure 1).

## 3. Materials and Methods

### 3.1. Measures and Questionnaire Development

The formulation of the formal questionnaire followed three steps: a pilot interview, the creation of measurement items, and a pre-survey. First, we designed an open interview questionnaire (see Table 1) to develop citizens’ beliefs towards participatory behaviors in fighting COVID-19, just as suggested by Francis et al. [87]. Then, twenty-six semi-structured pilot interviews were performed, recorded, and finally transcribed, in order to identify commonly held beliefs. Second, the operationalized items for the latent constructs in the TPB, the NAM, and POS were established by integrating the common themes elicited from the interviews with the statements in prior studies [40,46,87,88]. Third, three experts who are familiar with the TPB and the NAM and well-experienced in questionnaire design were invited to help modify the questionnaire. A pretest among 30 residents in the target population was then implemented to verify the face and content validity of the questionnaire. The final questionnaire, including seven latent variables and twenty-eight measurement items, was developed and measured on a seven-point Likert scale, with “1” indicating “strongly disagree” and “7” indicating “strongly agree”. The measurement items finally developed are shown in Table 2.

### 3.2. Data Collection

A cross-sectional anonymous online self-reported questionnaire survey was conducted in Harbin, the capital city of Heilongjiang province in northeast China. The reason that Harbin was chosen as the research area was mainly because the city experienced five major serious outbreaks of the COVID-19 pandemic during 2021, and the residents in Harbin participated extensively in pandemic prevention and control, which contributed greatly to each round of successful struggle against the disease. The target population of our research were residents who were over 16 years old. A question regarding whether the respondent was over 16 years was set at the beginning of the questionnaire to eliminate ineligible respondents.

To comply with the local preventive policies and maintain social distancing during the COVID-19 pandemic, we relied on a professional web-based survey platform called Tencent Questionnaire (https://wj.qq.com/, accessed on 12 May 2022) to collect the data. The online questionnaire was then distributed through WeChat, a widely used instant messaging tool in China, to urban residents in Harbin with the assistance of Northeastern University students and community workers. The survey was carried out from May to June in 2022; it lasted for about 30 days in total. There were 520 participants completing the online survey. Questionnaires finished in less than three minutes or with ten successive same answers or with more than two missing values were excluded to ensure the quality of the completed questionnaire for further analysis. Finally, 463 qualified questionnaires were used for the subsequent analysis, resulting in an overall effective response rate of 89.04%.

## 4. Results

### 4.1. Sample Profile

With regard to the demographic information of the sample, there were 54.6% males among the 463 respondents, which is slightly higher than the percentage of male residents in Harbin. A total of 40.2% of the respondents were under 35 years, and 28.7% of them were between 36 and 45 years old, whereas 24.8% of them were aged from 46 to 55. As for education, 33.3% of the respondents had high school diplomas, 18.8% were college graduates, and 5% had a master’s or doctoral degree. Concerning monthly income, 48.1% of the respondents reported income between CNY 2001 and CNY 5000, and 37.5% of them had a monthly income of less than CNY 2000. The distribution of survey participants’ occupations demonstrated that the respondents covered a variety of occupations, including civil servant, public institution employee, enterprise administrator, technician, industrial worker, commercial and service staff, non-profit staff, farmer, retiree, soldier, student, self-employed, and unemployed persons. Of the survey participants, 12.5% were members of Communist Party of China, 13.2% were members of the Communist Youth League of China, 0.9% were members of the democratic parties, and 73.4% were mass people. Apart from this, more than half of the respondents (65.5%) had lived in Harbin for more than six years, whereas 18.6% of them had resided there for less than three years.

### 4.2. Measurement Model Testing

To check the reliability and validity of the constructs and the fitness of the model, we first tested the measurement model with a confirmatory factor analysis. The analytical findings demonstrated that a suitable fitting result was achieved with the following parameters: χ^2^/df = 3.58, RMSEA = 0.075, CFI = 0.918, TLI = 0.905, SRMR = 0.047, which all satisfied the criteria recommended by Hu and Bentler [89].

The composite reliability and Cronbach’s alpha are regarded as reliability indexes for testing latent variables. As shown in Table 3, the values of composite reliability and Cronbach’s alpha of each construct were above 0.8, which were higher than the criteria of 0.6, thus reflecting the satisfactory intrinsic quality of the model. The average variance extracted (AVE) was tested to measure the convergent validity. The results showed that the AVE of all constructs was above 0.6, which was above the suggested threshold of 0.50 [90]. As for the content validity, all the factor loadings were higher than 0.7, which reflected that the contents of the measurement were suitable for the latent variables. The discriminant validity of our model was confirmed (see Table 4), because the AVE of each construct is higher than the squared correlations between the latent constructs, as suggested by Hair et al. [91].

### 4.3. Structural Model Testing

The maximum likelihood estimation method was used to analyze the theoretical model of influencing factors of citizens’ participatory behaviors in COVID-19 with structural equation modeling by Mplus 8.0. The goodness-of-fit statistics showed that the hypothesized model fit the data well: χ^2^ = 15.478, df = 6, RMSEA = 0.058, GFI = 0.994, CFI = 0.981, SRMR = 0.019. As displayed in Table 5, the hypotheses were all verified, and the model explained 62.9% of the variance in Chinese citizens’ participatory behavior against COVID-19.

First, we assessed the relationships among different variables in TPB. Specifically, ATT (β = 0.283, *p* < 0.001) and PBC (β = 0.279, *p* < 0.001) exerted a significant influence on citizens’ participatory behavior, and thus H1 and H2 were proved. The structural results also showed that both ATT (β = 0.146, *p* < 0.01) and SN (β = 0.255, *p* < 0.001) had a significantly positive relationship with PBC, hence confirming H3 and H5. The positive link between SN and ATT was also verified (β = 0.375, *p* < 0.001), and thus H4 was supported. Second, we tested the relationships among different variables in TPB and NAM. The hypothesis paths from AC (β = 0.323, *p* < 0.001) and SN to PN (β = 0.192, *p* < 0.001) were both supported by the empirical data, and H6 and H8 were consequently confirmed. In addition, the structural results also revealed that there was a significant link between AC and ATT (β = 0.268, *p* < 0.001), and therefore H7 was proved. Moreover, the analysis also demonstrated the expected positive effect of SN on PN (β = 0.192, *p* < 0.001), which validated H9. Finally, we examined the effects of OPP. Results of the structural model showed that OPP had a significant positive impact on ATT (β = 0.189, *p* < 0.01), PBC (β = 0.233, *p* < 0.001), and participation behavior (β = 0.087, *p* < 0.05). Therefore, H10, H11, and H12 were supported. Meanwhile, the testing model explained 45.4%, 56.7%, and 55.7% of the variance in PN, PBC, and ATT towards citizens’ participatory behavior, respectively. With respect to the total impacts, ATT had the most influence on citizens’ involvement in COVID-19 prevention and control, followed by PBC and OPP.

### 4.4. Mediation Analysis

The mediating effects within the variables were further examined using the bootstrapping method suggested by Mackinnon et al. [92]. The results displayed in Table 6 demonstrate that all the indirect effects were significant. Overall, PN, ATT, and PBC were the main mediators of the research model. Specifically, ATT mediated all the paths from AC (β = 0.076, *p* < 0.01), SN (β = 0.106, *p* < 0.001), PN (β = 0.108, *p* < 0.01), and OPP (β = 0.053, *p* < 0.05) to citizens’ participatory behavior, respectively. PBC mediated all the paths from ATT (β = 0.041, *p* < 0.01), SN (β = 0.071, *p* < 0.01), and OPP (β = 0.065, *p* < 0.01) to participation behavior. In addition, PN played a significant mediating role in the path from AC (β = 0.123, *p* < 0.01) and SN (β = 0.073, *p* < 0.01) to ATT. Moreover, ATT mediated the relationships between SN (β = 0.055, *p* < 0.01), PN (β = 0.056, *p* < 0.05), AC (β = 0.039, *p* < 0.05), and PBC. All these mediating effects reflected that the influencing mechanism of citizens’ participatory behavior in COVID-19 prevention and control was complicated.

## 5. Discussion and Implications

This study is a first attempt to investigate a variety of factors that influence citizens’ participation in COVID-19 prevention and control in China, based on an online survey of the residents in Harbin, which has been attacked by several waves of the COVID-19 pandemic since 2020. An integrative model of the TPB, the NAM, and POS were employed to test the hypothesized relationships among the three constructs of the original TPB model, two components of the NAM framework, and the OPP, which is the most frequently addressed dimension of the POS theory. A convenience sampling method based on a professional web-based survey platform was adopted to collect the cross-sectional data. The empirical results showed that the integrated model fit the data excellently and accounted for 62.9% of the variance in participation in COVID-19 prevention and control, which indicates the appropriateness of the comprehensive model in interpreting participation behaviors in the area of a public health emergency. In addition, the research findings of the present study not only have important theoretical contributions but also show some practical implications to inform policymakers in motivating citizens’ participation in COVID-19 prevention and control.

### 5.1. Theoretical Implications

Although previous research has estimated the social–psychological determinants of public participation in COVID-19 prevention and control, there have been no studies addressing this question in the Chinese context based on an integrated framework of both the TPB and NAM, to the best of our knowledge. Furthermore, no studies have simultaneously investigated social–psychological as well as external environmental factors that contribute to citizens’ participation in COVID-19 prevention and control, grounded in both social–psychological frameworks and established theories regarding environmental elements. The current study represents a first attempt to employ a combined theoretical framework of TPB, NAM, and POS in the context of Chinese citizens’ participation. The empirical analysis indicated that the integrative model possessed strong explanatory power and provided a comprehensive and effective framework to identify social–psychological and political environmental factors that influence citizens’ involvement in COVID-19 prevention and control. Therefore, the present study offers strong empirical evidence for integrating the components of TPB, NAM, and POS for an examination of the influential factors of pro-social behavior, including participation behavior, in the context of a public health emergency.

The POS theory has been widely adopted to interpret citizens’ diverse political activities. However, almost all the existing studies were aggregate-level analyses that have utilized national, regional, or local data [71,72]. Moreover, the research methods used were mainly the case study [72,79], media and interview content analysis [66,76], comparative analysis [74], and the ethnographic method [73]. No studies were found in extant research using quantitative analysis of individual-level data. In this paper, we tried for the first time to test POS theory by employing a structural equation model analysis of individual citizens’ survey data. Therefore, the results contribute to theory development by providing a quantitative measurement method for quantifying the dimensions of POS and using a new data analysis method based on individual-level data.

The analytical results of this study indicate that the degree of OPP not only significantly directly influenced Chinese citizens’ participation in COVID-19 prevention and control, but also significantly indirectly affected participation behaviors by way of ATT and PBC. This finding contributes to Zhang et al.’s and Nentwich’s [27,77] aggregate-level analyses that OPP plays an important role in stimulating public participation by adding individual-level empirical evidence. In addition, the results are also consistent with previous research stating that exogenous environmental factors markedly indirectly influence citizens’ preventative behaviors against the pandemic via the mediation of social-psychological factors [14,86]. This means that the degree of OPP could promote citizens’ participation in COVID-19 prevention and control by providing multiple ways of participation and opening channels and platforms for participation. It also indicates that the OPP, as outside political opportunities, would enhance citizens’ satisfaction towards the government’s policies, motivate their positive attitudes toward participation in COVID-19 prevention and control, and increase their confidence in and perceived capabilities of participation, thereafter stimulating participation behaviors. Furthermore, the structural results reveal that concerning the overall impact, the OPP was the third most influential factor that impacted citizens’ participation, closely following ATT and PBC. Previous studies have revealed the important role of the OPP but failed in investigating the extent of the effect. The estimation of this analysis reveals the degree of the influence of the OPP, making a further contribution to existing research.

ATT was found to be an important mediator in the analytical model, because ATT significantly mediated the impacts of the OPP, AC, PN, and SN on citizens’ participation behaviors. The important mediating role of ATT in explaining a person’s pro-social intentions/behaviors has been confirmed by a number of studies [36,40,57]. These research findings indicate that ATT is a key factor in influencing whether individuals would participate in the prevention and control of COVID-19. Other social–psychological and external environmental factors have influenced individuals’ participatory behaviors by changing their attitudes. In addition, ATT had the greatest effect with regard to the total impact. This finding corresponds with Zhang and Mu’s and Johnson and Hariharan’s arguments that ATT plays a critical role in affecting people’s mask-wearing behaviors [93,94]. It indicates that citizens are more likely to engage in pandemic prevention and control when they exhibit optimistic and positive attitudes.

As the results revealed, PBC is also an important mediator in the structural model, because it significantly mediates the effects of ATT, OPP, and SN on citizens’ involvement. This was consistent with the findings of multiple studies [14,36,38]. It implies that the OPP, ATT, and SN indirectly influenced participating behaviors by means of changing the level of one’s perceived behavior control. It is worth noting that ATT affects participation by way of PBC. This result is in accordance with Bamberg and Möser’s and Zhao’s findings [38,41]. It indicates that the positive or negative ATT would influence the level of citizens’ perceived capacities, beliefs, and self-confidence. Moreover, PBC was found to have the second largest total impact. This result complies with Das et al.’s and Fischer and Karl’s viewpoints that PBC is a profound antecedent of both COVID-19-relevant intentions and actual behaviors [9,10]. The analysis displays that individuals were more likely to behave in response to the COVID-19 pandemic when they felt that they had the ability to participate.

The findings indicate that SN was an important precedent of PN, ATT, and PBC. Such findings correspond with the analytical results of Zhang et al., López-Mosquera et al., and Bratt [57,95,96]. These results show that the expectations, approval, or encouragement from important persons and organizations, including the government, the community neighborhood committee, family members, friends, and role models, have effects on individuals’ evaluation about their behavior and the perceived easiness and moral correctness of participating in COVID-19 prevention and control.

With regard to the two constructs of the NAM, AC was significantly positively related with PN, which conforms with the original mediator model of NAM [50]. PN was found to indirectly impact participation behaviors through the mediation of attitudes, which violates the assumptions of the NAM that PN is a direct impact factor of behaviors. However, this finding is in agreement with the results of Klöckner and Wang et al. [61,62]. It illustrates that the feeling of personal moral obligation shapes people’s attitudes towards their behaviors. Citizens’ supportive moral norms, i.e., feeling they have an obligation to participate in COVID-19 prevention and control, leads to their positive and favorable evaluations of participation behaviors, which then motivates actual behaviors.

### 5.2. Practical Implications

Citizen participation is crucially important in the prevention and control of the COVID-19 and other public health emergencies. This study provides important practical implications for the Chinese government to effectively mobilize public participation in the context of the fight against epidemic diseases. First, the OPP is an important antecedent of citizens’ participation and has the third strongest overall impact. Therefore, it is necessary for the Chinese government to enhance its openness toward public participation mainly through releasing the latest information about the spread of the epidemic and relevant prevention and control measures in a timely manner, continuously expanding the channels and platforms for participation, providing diversified forms of participation, and improving policies regarding public participation based on citizens’ comments and suggestions.

Second, ATT and PBC both have the strongest total influence on citizens’ participation and play the most important mediating roles. Consequently, this study provides justifications for promoting citizens’ positive and favorable evaluations of epidemic prevention and control as well as citizens’ perceived capabilities to participate. The government should educate citizens that participation in epidemic prevention and control is a meaningful, effective, and cost-efficient way to constrain the disease from spreading and protect the health of citizens, by way of disclosing news reports and statistical data on the positive results of disease control. Various efforts, including accelerating openness to public participation, should also be made in order to make citizens’ participation easy, convenient, and economical, in order to increase citizens’ self-efficacy.

Third, because AC and SN are two important precedent factors that activate ATT, PN, and PBC, it is of vital importance to strengthen citizens’ awareness of the positive effects of participation in preventing and controlling epidemics by means of media campaigns, community interventions, school education, and government information disclosures. It is also essential to enhance citizens’ perceived degree of social pressures by building an atmosphere of social engagement in public health emergency management, selecting a few role models for citizens to follow, and increasing supports and encouragements from the government and community neighborhood committees. With the development of high AC and SN, Chinese citizens are more likely to form more favorable ATT, as well as higher levels of PBC and PN, which then motivates citizens’ participation behaviors.

## 6. Conclusions

Chinese citizens’ participation in the prevention and control of COVID-19 has contributed much to the successful fight against the pandemic. The factors influencing citizens’ participation behaviors have rarely been reported based on both social–psychological theories and political environment theory. This study is a pioneer that endeavors to explore various determinants of Chinese citizens’ participation in COVID-19 prevention and control by employing a combined analytical framework of TPB, NAM, and POS theory. The empirical results show the comprehensiveness, effectiveness, and high explanatory power of the postulated aggregative model. The outcomes of the current study also uncovered a few remarkable insights, such as the significantly direct and indirect impacts of the OPP, the important mediating roles and the large overall impacts of both ATT and PBC, the mediation of ATT between PBC and participation behaviors, the precedence of SN, and the significantly indirect effect of PN on citizens’ participation behaviors via ATT.

Although the present research contributes to theoretical development and provides some innovative insights into the literature of public health and public participation, there are still several possible limitations that also denote possibilities for future research. First, the questionnaire survey and cross-sectional data cannot be used to infer the causal relationship among different variables. More rigorous experimental designs should be used in the future to enhance the comprehension of the influencing factors of citizens’ participation in COVID-19 prevention and control. Second, the measurement of key variables is self-reported data, and we cannot deny that the respondents may over-report their participating behaviors, since these are a kind of prosocial behavior. Therefore, the measurement method should be further improved in the future. Third, all the respondents in this study were urban residents, who have higher education levels and better access to information than rural residents. Future studies could focus more on rural areas, which would provide further knowledge about the determinants of citizens’ participation in epidemic prevention and control. Finally, a convenience sampling technique based on the investigators’ networks was employed, which might lead to selection bias and limit the extrapolation of the results of this study to some extent. A random nation-wide sample could be used to validate the findings of this research in the future.

## Figures and Tables

**Figure 1 ijerph-19-15794-f001:**
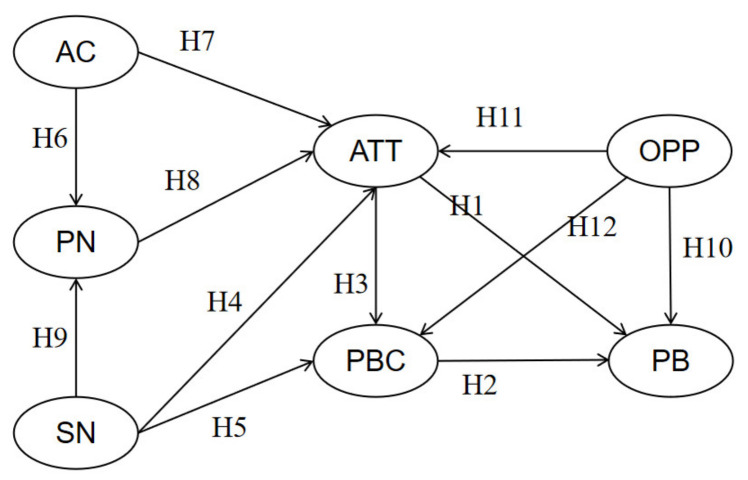
Integrated analytical model. Note: H = hypothesis.

**Table 1 ijerph-19-15794-t001:** Semi-structured pilot interview questions.

Semi-Structured Pilot Interview Questions
(a) What do you think to be the benefits or harms of participating in COVID-19 prevention and control for residents and the communities?
(b) What do you think to be the benefits or harms of participating in COVID-19 prevention and control for yourself?
(c) Are there any persons or organizations who would support your participating in COVID-19 prevention and control?
(d) What factors would make it easy or difficult for you to participate in COVID-19 prevention and control?
(e) What factors regarding the government’s openness to public participation would enable you or make it difficult for you to participate in COVID-19 prevention and control?

**Table 2 ijerph-19-15794-t002:** Operationalized items for the latent variables.

Operationalized Items for the Latent Variables
**Awareness of consequences (AC)**
AC1. Participation in COVID-19 prevention and control helps to protect citizens’ lives.
AC2. Participation in COVID-19 prevention and control helps to control the spread of the epidemic.
AC3. Participation in COVID-19 prevention and control helps to ensure community environmental health.
AC4. Participation in COVID-19 prevention and control helps to improve the efficiency of prevention and control and save social resources.
AC5. Participation in COVID-19 prevention and control helps to increase citizens’ knowledge and ability to fight against the epidemic.
**Attitudes toward participation in COVID-19 prevention and control (** **ATT)**
ATT1. It is meaningful to participate in COVID-19 prevention and control.
ATT2. It is a wise choice to participate in COVID-19 prevention and control.
ATT3. Participation in COVID-19 prevention and control is beneficial to my health.
ATT4. Participating in COVID-19 prevention and control is not a waste of my time and energy.
ATT5. It is worthwhile to participate in COVID-19 prevention and control to protect my health.
**Subjective norms (SN)**
SN1. My family and my friends would be in favor of my participation in COVID-19 prevention and control.
SN2. Most people like me have participated in COVID-19 prevention and control.
SN3. The government and the community neighborhood committee would support my participation in COVID-19 prevention.
SN4. My role models would approve of my participation in COVID-19 prevention and control.
**Degree of openness to public participation (OPP)**
OPP1. I think the policies regarding public participation in COVID-19 prevention and control are satisfactory.
OPP2. I think the policies regarding public participation in COVID-19 prevention and control are conducive to citizens’ participation.
OPP3. I think the information disclosure about COVID-19 prevention and control is timely and transparent.
OPP4. I think the channels and platforms for participation are open for citizens to participate in COVID-19 prevention and control.
OPP5. I think there are various forms of participation for citizens to participate in COVID-19 prevention and control.
**Perceived behavioral control (PBC)**
PBC1. I believe that I am able to participate in COVID-19 prevention and control even if I do not have plenty of resources, time, and opportunities.
PBC2. I am confident that if I want, I can participate in COVID-19 prevention and control.
PBC3. I am sure that I can overcome various difficulties to participate in COVID-19 prevention and control.
**Personal norms (PN)**
PN1. It is my moral obligation to participate in COVID-19 prevention and control.
PN2. I would feel guilty for not participating in COVID-19 prevention and control.
PN3. I feel that I should try my best to participate in COVID-19 prevention and control.
**Participation behaviors (PB)**
PB1. I often learn about COVID-19 prevention and control through various channels.
PB2. I participate in COVID-19 prevention and control in various ways.
PB3. I actively participate in COVID-19 prevention and control.

**Table 3 ijerph-19-15794-t003:** Results of confirmatory factor analysis.

Latent Variable	Observation Variable	Factor Loadings	R^2^	Cronbach’s α	AVE	CR
AC	AC1	0.78	0.608	0.883	0.6043	0.8837
	AC2	0.841	0.708			
	AC3	0.821	0.674			
	AC4	0.699	0.488			
	AC5	0.737	0.544			
PN	PN1	0.779	0.607	0.837	0.6297	0.836
	PN2	0.781	0.61			
	PN3	0.82	0.673			
SN	SN1	0.828	0.685	0.887	0.6658	0.8883
	SN2	0.861	0.742			
	SN3	0.802	0.643			
	SN4	0.77	0.592			
OPP	OPP1	0.832	0.692	0.922	0.7102	0.9243
	OPP2	0.841	0.708			
	OPP3	0.894	0.799			
	OPP4	0.887	0.787			
	OPP5	0.752	0.566			
PBC	PBC1	0.804	0.647	0.865	0.6804	0.8646
	PBC2	0.846	0.715			
	PBC3	0.824	0.679			
ATT	ATT1	0.846	0.715	0.931	0.7352	0.9326
	ATT2	0.904	0.818			
	ATT3	0.922	0.85			
	ATT4	0.773	0.598			
	ATT5	0.834	0.696			
PB	PB1	0.821	0.674	0.860	0.6731	0.8606
	PB2	0.836	0.698			
	PB3	0.804	0.646			

Notes: R^2^ denotes reliability coefficient; CR denotes composite reliability; AVE denotes average variance extracted.

**Table 4 ijerph-19-15794-t004:** Convergent and discriminant validity.

Variable	AC	PN	SN	OPP	PBC	ATT	PB
AC	0.6043						
PN	0.5013	0.6297					
SN	0.1849	0.3422	0.6658				
OPP	0.2480	0.2970	0.4816	0.7102			
PBC	0.1616	0.2938	0.5580	0.5461	0.6804		
ATT	0.3636	0.4489	0.4409	0.3931	0.4382	0.7352	
PB	0.2570	0.3516	0.5112	0.4692	0.6448	0.6241	0.6731

Notes: The numbers in diagonal are the average variance extracted by each variable. The numbers below the diagonal are the squared correlation coefficients between the variables.

**Table 5 ijerph-19-15794-t005:** Standardized path coefficients of the structural model.

Hypothesis	PATH	Estimate	S.E.	Est./S.E.	*p*-Value	Result
H1	ATT→PB	0.283 ***	0.048	5.898	0.000	Supported
H2	PBC→PB	0.279 ***	0.064	4.343	0.000	Supported
H3	ATT→PBC	0.146 **	0.052	2.786	0.005	Supported
H4	SN→ATT	0.375 ***	0.094	3.995	0.000	Supported
H5	SN→PBC	0.255 ***	0.070	3.652	0.000	Supported
H6	AC→PN	0.323 ***	0.038	8.576	0.000	Supported
H7	AC→ATT	0.268 ***	0.070	3.842	0.000	Supported
H8	PN→ATT	0.382 ***	0.118	3.233	0.001	Supported
H9	SN→PN	0.192 ***	0.037	5.154	0.000	Supported
H10	OPP→PB	0.087 *	0.039	2.228	0.026	Supported
H11	OPP→ATT	0.189 **	0.069	2.744	0.006	Supported
H12	OPP→PBC	0.233 ***	0.053	4.383	0.000	Supported
Total variance explained: R^2^ of PN = 0.454 R^2^ of PBC = 0.567 R^2^ of ATT = 0.557 R^2^ of PB = 0.629			Standardized total impact on participating behavior: AC = 0.076 PN = 0.108 SN = 0.177 ATT = 0.324 PBC = 0.279 OPP = 0.205			

Notes: *** *p* < 0.001; ** *p* < 0.01; * *p* < 0.05.

**Table 6 ijerph-19-15794-t006:** Results of mediation analysis.

Variable	Mediator	Variable	Standardized Estimate	Standard Error	*p*-Value
AC	ATT	PB	0.076 **	0.025	0.002
SN	ATT	PB	0.106 ***	0.024	0.000
PN	ATT	PB	0.108 **	0.037	0.004
OPP	ATT	PB	0.053 *	0.025	0.033
SN	PBC	PB	0.071 **	0.026	0.006
ATT	PBC	PB	0.041 **	0.016	0.009
OPP	PBC	PB	0.065 **	0.023	0.005
AC	PN	ATT	0.123 **	0.044	0.005
SN	PN	ATT	0.073 **	0.027	0.006
SN	ATT	PBC	0.055 **	0.019	0.004
PN	ATT	PBC	0.056 *	0.028	0.049
AC	ATT	PBC	0.039 *	0.018	0.032

Notes: *** *p* < 0.001; ** *p* < 0.01; * *p* < 0.05.

## Data Availability

The data presented in this study are available in [the Appendix A].

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
