# Peer review of "Factors Contributing to Citizens’ Participation in COVID-19 Prevention and Control in China: An Integrated Model Based on Theory of Planned Behavior, Norm Activation Model, and Political Opportunity Structure Theory"

_ijerph, 2022, doi:10.3390/ijerph192315794_

Round 1
Reviewer 1 Report
Thank you for asking me to review this article. The theme chosen by the authors is certainly worthy of attention especially in consideration of the fact that, with the SARS-CoV-2 pandemic, the planned behavior has assumed an important role in public health. The pandemic context in progress, in fact, has highlighted how the mere information on the correct behavior to control the spread of infections is not a sufficient condition to contain the infections: it is essential to educate the general population to adopt them. Seemingly simple decisions, such as respecting isolation, washing your hands frequently, wearing masks, not touching your face when on the street, or disinfecting items that could be contaminated, in the context of the pre-vaccine COVID-19 pandemic have certainly had a decisive incidence and often a life-saving role.
The authors aim to explore the determinants of Chinese citizens' participation in the prevention and control of COVID-19 based on a combined model of the Theory of planned Behavior, norm activation model and political opportunity structure theory. The chosen theme, therefore, is certainly very interesting and has been deepened with an accurate description. However, the form that has been given to the manuscript is distant from the characteristic structure of a scientific article and seems to fit better in a chapter of a book. The concepts are certainly well described and present in detail the context in which the research question develops. The presentation of the contents, however, is not very concise and distracts the reader's attention from the focus on which the survey is elaborated. Since this is a survey, I advise the authors to dwell on the topic under consideration to highlight it, reducing the space dedicated to the premises and leaving considerations and food for thought in the discussion section.
The methods are described in a discursive but not completely exhaustive manner; the authors have implemented a survey administered online but do not clarify the method of administration (questionnaire sent by email or through the use of a dedicated platform, etc.), including the time dedicated to each of the interviewees; in fact, the consequentiality of the questions proposed, the time necessary to provide the answers and the methods of recruitment and administration can positively or negatively influence the choice of the answer; therefore a precise outline could better clarify these aspects, make the reading of the manuscript more fluent and open the discussion to other food for thought.
Furthermore, in the paragraph “3.2 Data Collection and Sample Profile” lines 289-296, the authors anticipate the demographic distribution of the sample. Unless the sample was selected by the authors upstream of the administration (as an inclusion criterion), in my opinion it would be more appropriate to include these contents in the results paragraph. However, since in literature it is possible to see this section described in the methods, the authors may consider not following this suggestion. I believe that, by paying attention to these suggestions, the manuscript can represent an interesting contribution to the academic literature.
Reviewer 2 Report
The paper deals with a very interesting and current topic, and therefore the paper could be of great interest to the readers.
My suggestions refer to two main aspects of the paper:
1. The term citizen/public participation in the COVID-19 prevention and control needs to be elaborated, since it is not clear (not from the introduction and neither from theoretical section) what does the participation encompass. Which types or instruments of participation authors refer to? The title suggests the paper will investigate specific types of participation related to Covid-19 policy, however this is not explained and it appears that authors use the term participation as referring to adherence to Covid-19 measures. If so, this has to be more precisely explained. The term participation is quite broad and can embrace different modes of citizens' involvement in different phases of formulating or implementing public policy.
2. The methodological part is explained is detail, however, since the survey participants are self-selected the sample is not representative, which is why this should be stressed when discussing the results and their implications.
Round 2
Reviewer 1 Report
The authors have made all the suggested changes to the manuscript and, in my opinion, the quality of the manuscript has greatly improved. I therefore believe that the manuscript can be published in its present form.